# Prognostic Factors for Survival in Glioblastoma: A Retrospective Analysis Focused on the Role of Hemoglobin

**DOI:** 10.3390/biomedicines12061210

**Published:** 2024-05-29

**Authors:** Zuzana Pleskacova, Michael Bartos, Hana Vosmikova, Rafael Dolezal, Petr Krupa, Barbora Vitovcova, Petra Kasparova, Emil Rudolf, Veronika Skarkova, Denisa Pohankova, Veronika Novotna, Jiri Petera

**Affiliations:** 1Department of Oncology and Radiotherapy, University Hospital and Medical Faculty, 500 05 Hradec Kralove, Czech Republic; denisa.pohankova@fnhk.cz (D.P.); veronika.bockayova@fnhk.cz (V.N.); jiri.petera@fnhk.cz (J.P.); 2Department of Neurosurgery, University Hospital and Medical Faculty, 500 05 Hradec Kralove, Czech Republic; michael.bartos2@fnhk.cz (M.B.); petr.krupa@fnhk.cz (P.K.); 3The Fingerland Department of Pathology, University Hospital and Medical Faculty, 500 05 Hradec Kralove, Czech Republic; hana.vosmikova@fnhk.cz (H.V.); petra.kasparova@fnhk.cz (P.K.); 4Biomedical Research Centre, University hospital and Medical Faculty, 500 05 Hradec Kralove, Czech Republic; rafael.dolezal@fnhk.cz; 5Department of Medical Biology and Genetics, Medical Faculty, 500 05 Hradec Kralove, Czech Republic; vitovcob@lfhk.cuni.cz (B.V.); rudolf.emil@lfhk.cuni.cz (E.R.); hanusovav@lfhk.cuni.cz (V.S.)

**Keywords:** glioblastoma, prognostic markers, HIF, hemoglobin

## Abstract

*Background*: Although several prognostic factors for survival have been identified in glioblastoma, there are numerous other potential markers (such as hemoglobin) whose role has not yet been confirmed. The aim of this study was to evaluate a wide range of potential prognostic factors, including HIF-1α and hemoglobin levels, for survival in glioblastoma. A secondary aim was to determine whether hemoglobin levels were associated with HIF-1α expression. *Methods:* A retrospective study of 136 patients treated for glioblastoma at our institution between 2012 and 2021 was performed. Cox univariate and multivariate analyses were carried out. Kaplan–Meier survival curves were generated. In addition, bivariate non-parametric correlation analyses were performed for key variables. *Results:* Median survival was 11.9 months (range: 0–119.4). According to the univariate analysis, 13 variables were significantly associated with survival: age, performance status, extent of surgery, tumor depth, tumor size, epilepsy, postoperative chemoradiotherapy, IDH mutations, CD44, HIF-1α, HIF-1β, vimentin, and PDFGR. According to the multivariate regression analysis, only four variables remained significantly associated with survival: age, extent of surgery, epilepsy, and HIF-1α expression. No significant association was observed between hemoglobin levels (low <120 g/L in females or <140 g/L in males vs. high ≥120 or ≥140 g/L) and survival or HIF-1α/HIF-1β expression. *Conclusions:* In this retrospective study of patients with glioblastoma, four variables—age, extent of surgery, HIF-1α expression, and epilepsy—were significant prognostic factors for survival. Hemoglobin levels were not significantly associated with survival or HIF-1α expression. Although hypoxia is a well-recognized component of the glioblastoma microenvironment, more research is needed to understand the pathogenesis of onset tumor hypoxia and treatment implication.

## 1. Introduction

Glioblastoma, the most common type of malignant brain tumor [1], is characterized by intratumoral heterogeneity, an invasive growth pattern, and poor response to treatment [2]. In the last two decades, the most significant advance in the treatment of glioblastoma was the addition of postoperative chemoradiation (Stupp’s protocol) [3]. Survival outcomes are still poor, with a median survival of approximately 15 months [4] and 5-year survival rates of around 5% [2].

Confirmed prognostic factors for survival in glioblastoma included performance status, age, extent of resection, postoperative chemoradiotherapy, MGMT methylation, and IDH (isocitrate dehydrogenase) 1/2 mutation [5]. In addition, several other potential prognostic factors have been identified, including hemoglobin level [6,7], epilepsy [8,9], and subventricular zone involvement [10]. However, the true prognostic value of these parameters is uncertain due to the inconsistent results reported to date.

Tumor hypoxia is a common feature in glioblastoma, mainly due to abnormal neovascularization. Hypoxic stress triggers the accumulation of hypoxia-inducible factor 1 alfa (HIF-1α) and its partner HIF beta (HIF-1β), which initiate the transcription of hundreds of genes involved in the regulation of angiogenesis, glycolysis, autophagy, motility and invasion, and resistance to chemotherapy and radiation. HIF-1α expression is associated with poor prognosis in gliomas [11]. The low hemoglobin level before radiotherapy was proved to be the negative prognostic factor of survival of glioblastoma [7], but this finding is not consistent across all studies. A decrease in hemoglobin levels can contribute to the upregulation of HIF expression in glioblastoma, further promoting tumor growth and aggressiveness. Understanding the interplay between hemoglobin levels and HIF expression in glioblastoma may provide valuable insights into potential therapeutic strategies to target hypoxia and improve patient outcomes. Although there are some publications dealing with hemoglobin level and HIF expression in several tumor types [12,13], publications on glioblastoma are lacking.

In this context, the aim of the present study was to evaluate clinical and molecular factors—particularly HIF-1α and hemoglobin levels—to determine whether these variables are associated with survival in patients with glioblastoma. A secondary aim was to determine whether hemoglobin levels were associated with HIF-1α expression.

## 2. Patients and Methods

### 2.1. Patients

Patients diagnosed and treated for glioblastoma between 2012 and 2021 at our hospital (Hospital Hradec Kralove, Czech Republic) were evaluated for possible study inclusion; 324 patients were treated during the study period. Of these, 136 had complete clinical data, including tissue samples.

The patients’ demographic and clinical variables were obtained from clinical records, as follows: age, sex, data of diagnosis, type of procedure (gross total resection [GTR], subtotal resection [STR], or biopsy alone), clinical course, and imaging, laboratory, and histopathological data. Other data obtained from the medical records included: comorbid medical conditions, Karnofsky performance status (KPS), epilepsy (yes/no), tumor size, and hemoglobin levels.

Table 1 shows the main characteristics of the cohort. The cohort included 136 patients (94 males, 42 females). The median age was 59 years (range: 26–83). The type of surgery was distributed as follows: GTR (*n* = 54; 39.7%), STR (*n* = 64; 47.1%), and biopsy alone (*n* = 8; 13.2%). In nearly all cases (*n* = 135; 99.3%), the tumor was located in the supratentorial region. Of these, 125 (92%) were located in the superficial area (cortical or subcortical area) and 10 (7.4%) in deep anatomical structures (basal ganglia and/or corpus callosum). At diagnosis, 18 patients (13.2%) presented multifocal spread.

Ninety-three patients (68.4%) received standard temozolomide (75/mg^2^) administered concomitantly with conventional radiation therapy (maximum dose, 60 Gy). Palliative radiotherapy was performed in 31 patients (22.8%). The patients (*n* = 12, 8.8%) with an ECOG (Eastern Cooperative Oncology Group) performance status >3 received best supportive care.

Patients were stratified according to the peak preoperative hemoglobin levels. Hemoglobin levels were classified into low (<120 g/L in females, <140 g/L in males) and high (≥120 or ≥140 g/L, respectively).

### 2.2. Histochemical Analyses

All histological samples were formalin-fixed and paraffin-embedded following standard procedures at the hospital laboratory. Review of all available archival material, including immunohistochemical slides, was performed by a single experienced neuropathologist prior to study inclusion. Tissue blocks were cut into 2-mm-thick sections for immunohistochemical analysist.

We reviewed the literature to identify histochemical parameters potentially associated with hypoxia signaling [14,15,16], which included the following: IDH, CD44 (family of transmembrane glycoproteins), HIF-1α, HIF–1β, platelet-derived growth factor receptors (PDFGR), p16, p53, vimentin (VIM), SRY-Box Transcription Factor 11 (SOX11), D240 (podoplanin), MER proto-oncogene tyrosine kinase (MERTK), mesenchyme Homeobox 2 (MEOX2), oligodendrocyte transcription factor 2 (OLIG2), *signal transducer and activator of transcription 3 (STAT3)*, epidermal growth factor receptor (EGFR), CD34 (transmembrane phosphoglycoprotein), and *glial fibrillary acidic protein (GFAP)*.

Staining was carried out with the Ventana Benchmark Ultra immunostainer (Ventana/Roche, Tucson, AZ, USA) or the DAKO Omnis stainer (Agilent, Santa Clara, CA, USA). The Ventana OptiView DAB IHC kit (or DAKO EnVision Flex kit) was used for visualization: both methods use an avidin–biotin complex method with horseradish peroxidase as an enzyme and DAB (3,3′-diaminobenzidine) as chromogen. All slides were counterstained with hematoxylin, and positive controls were used on the slides, as appropriate.

The immunohistochemistry results were assessed by a single observer (HIF-1α, HIF-1β, STAT3) or two observers (remaining markers) blinded to the clinical and pathological data. First, the observer determined the percentage of positive tumor cells and staining intensity, which was rated on a scale from 1 to 3 (mild, moderate, or strong). Tumors in which <1% of cells were positive were considered negative (Figure 1). A modified H-score was then calculated by multiplying the most common staining intensity by the percentage of positive cells (range: 0 to 300). Consensus meeting with both observers was held for discrepant cases to attain a consensus score.

### 2.3. Statistical Analysis

The following variables were evaluated to check for an association with survival outcomes: age at first surgery, sex, KPS at diagnosis, surgery (radical vs. nonradical), hemoglobin levels at surgery, tumor localization (deep vs. superficial), tumor (T) size (<10 vs. ≥10 mm), epilepsy (yes vs. no), postoperative radiochemotherapy (yes/no), and the mutation status of the following: IDH, CD44, HIF-1α, HIF–1β, p53, p16, PDFGR, VIM, SOX11, D240, MERTK, MEOX2, OLIG2, EGFR, STAT3, CD34, and GFAP.

Survival was calculated as the time from surgery to date of death. For censored cases, survival was defined as the time from surgery until the last medical visit.

Survival data were analyzed using the Kaplan–Meier method with 95% confidence intervals (CI). The statistical significance of the Kaplan–Meier models was evaluated by log-rank Mantel–Cox (MC) and Tarone–Ware (TW) tests of the overall model *χ*^2^. All variables were tested in a univariate Cox regression model and in multivariate Cox models. The overall statistical significance of the Cox models was estimated from the total *χ*^2^. Identifiability of the models was monitored by controlling the maximal values in the correlation matrix of the regression coefficients. The top-scoring statistical models were tested by random bootstrapping (1000 repetitions) to estimate model robustness or bias-corrected accelerated CIs. Only the statistically significant variables (type I error, *p* < 0.05) were considered when interpreting the results. Bivariate non-parametric correlation analyses were carried out (Spearman *ρ* and Kendal *τ* criteria) to assess the mutual independence of the variables: epilepsy and tumor size, radicality of resection, HIF-1α, STAT3, IDH, and age. All statistical analyses were performed with the IBM-SPSS Statistical software program (v. 29.0).

## 3. Results

Median survival in the full cohort (*n* = 136) was 17.6 months (range: 0–119.4). The overall survival (OS) rate at one and three years was 47.8% (*n* = 65) and 12.5% (*n* = 17), respectively. Currently (as of 8 March 2024), four patients remain alive (60.8, 31.8, 38.0, and 38.8 months from surgery, respectively).

Survival was significantly correlated (*p* < 0.05; Kaplan–Meier) with the following variables: age (<70 vs. ≥70 years), KPS (≥70 vs. <70), extent of surgery (radical vs. nonradical), tumor depth (superficial vs. invasion of central structures), tumor size (<10 vs. ≥10 mm), epilepsy (yes vs. no), postoperative chemoradiotherapy (yes vs. no), and IDH mutation (yes vs. no) (Table 2, Figure 2).

On the Cox univariate analyses, survival was significantly (*p* < 0.05) associated with the following variables: age, KPS, extent of surgery, tumor depth, tumor size, epilepsy, postoperative chemoradiotherapy, IDH mutations, CD44, HIF-1α, HIF-1β, vimentin, and PDFGF. Since the HIF-1α marker is a well-known predictor of OS in patients with GBM, a histogram showing the distribution of the HIF-1α levels for short and long OS revealed in this study is provided in Figure 3. None of the other variables involved in this study were found statistically significant.

On the multivariate Cox regression analysis only age, extent of surgery, epilepsy, and HIF-1α expression were significantly associated with survival (Table 3).

Hemoglobin levels (low vs. high) did not correlate with survival of HIF-1α or HIF-1β expression (Kaplan–Meier).

On the bivariate analysis between HIF-1α and p53 and p16, neither the Spearman ρ nor the Kendal τ coefficients were statistically significant.

We tested possible correlation between epilepsy with tumor size, extent of surgery, HIF-1α, and IDH using a non-parametrical Spearman *ρ* model. A younger age (under 60 years), smaller tumors before treatment (<10 mm), nonradical surgery, and IDH mutations were statistically significant factors. Additional Cox multivariate analyses can be found in Appendix A.

## 4. Discussion

In this retrospective study, we evaluated numerous clinical and molecular factors—including HIF-1α and hemoglobin levels—to determine their potential association with survival in patients with glioblastoma. On the univariate analysis, numerous variables were significantly associated with survival. However, on the multivariate analysis, only four variables—age, extent of surgery, epilepsy, and HIF-1α expression—remained significant. There was no correlation between hemoglobin levels and survival or HIF-1α expression, which suggests that hemoglobin is not a prognostic factor for survival in this patient population.

In glioblastoma, hypoxia is closely associated with tumor aggressiveness, radiation resistance, chemoresistance, and poor prognosis [14]. Several studies have assessed the role of hemoglobin levels in patients with glioblastoma, with contradictory findings. Céfaro et al. [7] found that hemoglobin levels ≤12 g/L (versus > 12 g/L) was associated with significantly worse survival outcomes (12 vs. 23 months). Lutterbach et al. [5] also found that low hemoglobin levels were significantly associated with poor treatment outcomes. In a previous study [17], our group found that hemoglobin levels >12 g/L were associated with longer median survival compared to levels ≤12 g/L (12.1 vs. 7.9 months, respectively). More recently (2020), Kismar-Elbaz et al. [6] found that low hemoglobin levels (<12 g/dL [females] or <14 g/dL [males]) was a negative prognostic factor for survival, leading the authors to conclude that measures should be taken to normalize preoperative hemoglobin levels and red cell distribution width to improve prognosis. Notwithstanding the findings of those studies, other studies have not found any association between hemoglobin levels and prognosis in these patients [18,19]. In our previous study [17], we hypothesized that low levels of hemoglobin would increase HIF expression. However, in the present study, hemoglobin levels had no impact on HIF expression, nor were they associated with survival outcomes.

The mechanism of action of anemia on treatment outcomes in patients with glioblastoma remains unclear. In fact, only a few studies have assessed the influence of anemia on HIF expression. Winter et al. [12] evaluated HIF-1α expression in surgically-treated head and neck cancer patients and found that high HIF-1α/HIF-2α expression was an independent negative prognostic factor for survival. However, no correlation was observed between HIF-1α or HIF-2α expression and hemoglobin levels, leading those authors to conclude that activation of HIF and its signaling pathways is independent of hemoglobin levels, including hypoxia-independent mechanisms. On the other hand, Dallas et al. [13] found that patients with cervical cancer treated by radiotherapy and hemoglobin levels <11 g/dL showed elevated HIF-1α expression compared to patients with hemoglobin levels >12.5 g/L (*p* = 0.04).

A meta-analysis carried out to assess the role of HIF expression in glioblastoma found that HIF-1α expression was associated with high grade glioma and worse survival [11]. Sfifou et al. [20] recently found that 12-month survival outcomes were better in patients with negative HIF-1α expression and positive IDH1 when compared to patients with HIF-1α-positive, IDH1-negative disease. Potharaji et al. [21] found that strong HIF-1α expression was an independent prognostic factor for poor survival. In that study, the subgroup with the worst prognosis had strong expression of both HIF-1α and telomerase reverse transcriptase. In our study, HIF-1α expression was a strong, independent predictor of shorter survival. HIF transcription factors constitute the master regulators of the hypoxia adaptive response. Hypoxia occurs in GBM due to increased cell proliferation and tumor growth resulting in tumor neovascularization, which is insufficient and leads to hypoxia, acidosis, necrosis and intratumoral edema. Some genetic alteration (activation of epidermal growth factor receptor, the loss of tumor suppressor function—p53, PTEN) can increase HIF expression. Hypoxia is a main promotor of glioblastoma invasion, radio- and chemoresistace, and survival [22]. Further research on HIF factor expression regulations is a key factor for the development of targeted biological therapy directed at HIF [23].

Although our study showed a significant association between epilepsy and better survival, the true prognostic value of epilepsy remains unclear due to the heterogenous results reported in previous studies. For example, Dobran et al. [9] found that epilepsy at presentation was an independent prognostic factor for longer survival, but only in patients younger than 60 years. In the study by Berendsen et al. [8], epilepsy was positively correlated with longer survival. Interestingly, there were no survival differences between patients with or without antiepileptic medication, which suggests that the medication had no impact on survival. In a more recent study by that same research group [24], epilepsy was an independent prognostic factor for better survival in glioblastoma patients. In that study, epileptogenic glioblastoma was correlated with decreased hypoxia/HIF-1α/STAT5b signaling compared to non-epileptogenic glioblastomas. In our study, epilepsy was significantly correlated with tumor size (i.e., smaller tumors were associated with a higher probability of epilepsy), younger age, nonradical surgery, and IDH mutations. Other authors have also found a relationship between epilepsy and IDH mutations [25,26]. By contrast, no such association was observed in the study by Berendesen et al. [8].

Although we evaluated numerous different histochemical parameters potentially linked with hypoxia signaling, glioblastoma invasiveness, and disease progression, the only histochemical variable that was significantly associated with survival on the multivariate analysis was HIF-1α expression.

### Strengths and Limitations

The main limitation of this study is its retrospective study design. By contrast, an important strength is its large sample size.

## 5. Conclusions

In this retrospective study, four variables—age, extent of surgery, expression of HIF-1α, and epilepsy—were significant prognostic factors for survival in patients with glioblastoma. Importantly, hemoglobin levels were not correlated with survival or HIF-1α expression. Commonly used treatment modalities to correct anemia in cancer patients improve the quality in life and may improve their prognosis and response to treatment particularly when radiation therapy is planned. However, there is still no strong evidence that hemoglobin level in glioblastoma patient have an impact on survival. It may be caused by a higher influence of other prognostic markers, be they clinical (age, extent of surgery) or molecular (HIF-1α, IDH), or the tumor microenvironment. Although hypoxia is a well-recognized component of the glioblastoma microenvironment, more research is needed to better understand the pathogenesis of its onset and the treatment implications.

## Figures and Tables

**Figure 1 biomedicines-12-01210-f001:**
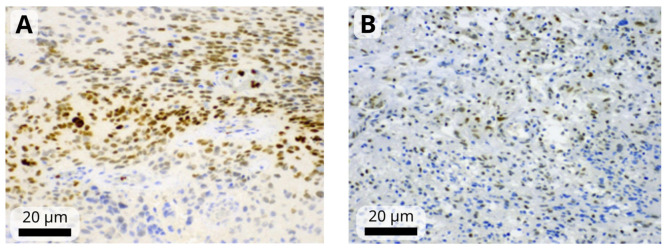
Representative images of moderate-to-strong nuclear immunoreactivity of HIF-1α (**A**) and weak-to-moderate immunoreactivity of HIF-1β (**B**) in glioblastoma cells. Images were obtained at magnification of 200×. The black scale bars indicate 20 µm.

**Figure 2 biomedicines-12-01210-f002:**
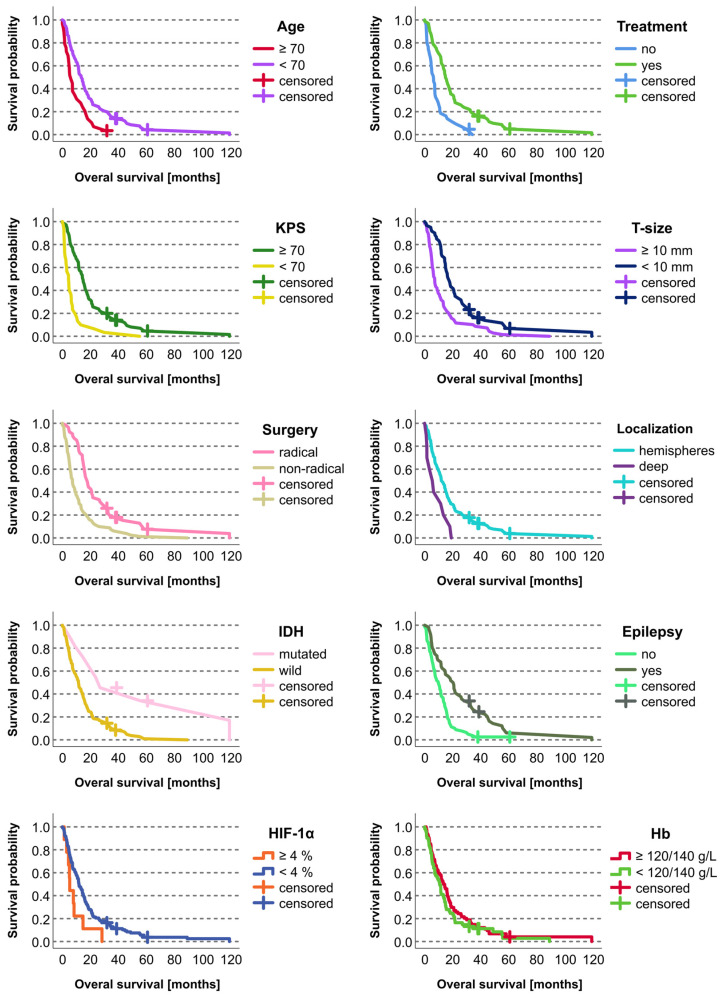
Kaplan–Meier curves for statistically significant categorical prognostic factors in the studied cohort of 136 patients with glioblastoma. The retrospective cohort included 132 deceased patients and four who were alive at the time of the study evaluation. The living patients were included as censored cases.

**Figure 3 biomedicines-12-01210-f003:**
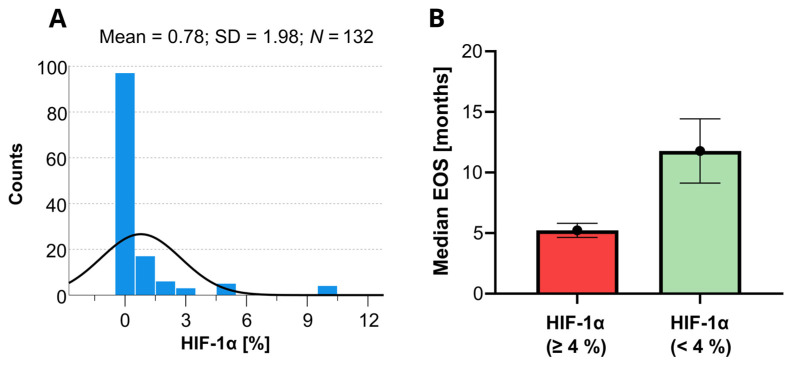
Histogram of the available HIF-1α data determined within the present study (**A**) and the corresponding expected median overall survivals (EOS) with the 95% confidence limits as predicted by the Kaplan–Meyer analysis (**B**). The black line in the histogram represents the expected normal distribution.

**Table 1 biomedicines-12-01210-t001:** Basic statistical characteristics of the involved patients.

Variable	Number of Patients (%)
**Total**	136 (100%)
**Sex**	
Male	94 (69.1%)
Female	42 (30.9%)
**Age (mean 59.3 year)**	
<70 year	110 (80.9%)
≥70 year	26 (19.1%)
**KPS, mean**	
≥70	87 (64%)
<70	49 (36%)
**Surgery**	
GTR	54 (39.7%)
STR	64 (47.1%)
Biopsy alone	18 (13.2%)
**Peak hemoglobin level ***	
Low	62 (45.6%)
High	74 (54.4%)
Epilepsy	
Yes	56 (41.1%)
No	80 (58.8%)
**Localization ****	
Superficial	125 (92%)
Deep	11(8%)
**Postoperative treatment**	
Chemoradiotherapy	93 (68%)
Radiotherapy	31 (23%)
Best supportive care	12 (9%)

**Abbreviations:** KPS: Karnofsky performance score; GTR: gross total resection (radical surgery); STR: subtotal resection. * The cut-off for low vs. high hemoglobin levels in females was <120 g/L vs. ≥120 g/L. In males, the cut-off was <140 g/L vs. ≥140 g/L. ** Superficial localization = cortical or subcortical area; deep localization = deep anatomical structures (basal ganglia or corpus callosum).

**Table 2 biomedicines-12-01210-t002:** Results of the Kaplan–Meier analyses.

Factor	*Subclass*	*N_C_*	*N_E_*	EOS_med_ [Months]	^0.95^CI [Months]	_MC_*χ*^2^ (*p*)	_TW_*χ*^2^ (*p*)
**Age**	<70 year	3	104	13.630	10.810–16.450	11.169 (<0.001)	12.057 (<0.001)
	≥70 year	1	28	6.130	2.438–9.822		
**Sex**	Male	4	90	11.530	7.986–15.074	0.044 (0.833)	0.039 (0.843)
	Female	0	42	11.930	8.578–15.282		
**KPS**	≥70	4	101	14.730	12.638–16.822	28.788 (<0.001)	37.463 (<0.001)
	<70	0	30	4.370	2.357–6.383		
**Surgery ***	Radical	4	50	17.570	14.689–20.451	21.940 (<0.001)	26.693 (<0.001)
	Non-radical	0	80	11.770	9.453–14.087		
**Hb ****	Low	3	58	10.770	7.851–13.689	1.437 (0.231)	1.711 (0.191)
	High	1	73	13.370	9.855–16.885		
**Location**	Peritumoral	4	120	11.930	9.202–14.658	5.850 (0.016)	5.514 (0.019)
	Deep	0	10	5.470	0.775–10.165		
**T-size**	<10 mm	4	56	16.930	14.197–19.663	17.757 (<0.001)	22.673 (<0.001)
	≥10 mm	0	69	7.230	5.638–8.822		
**Epilepsy**	Yes	2	54	20.300	15.411–25.189	20.968 (<0.001)	20.071 (<0.001)
	No	2	77	9.830	6.704–12.956		
**Treatment**	Yes	3	90	14.900	13.036–16.764	24.995 (<0.001)	27.615 (<0.001)
	No	1	42	6.370	3.672		
**IDH**	Mutated	2	9	26.900	0.000–63.481	12.406 (<0.001)	9.324 (<0.002)
	Wild	2	122	11.500	8.951–14.049		
**HIF-1α**	≥4	0	9	5.230	4.646–5.814	4.241 (0.039)	4.076 (0.043)
	<4	3	119	11.770	9.118–14.422		

**Abbreviations:** Hb: hemoglobin; KPS: Karnofsky performance status; EOS: estimated overall median survival; ^0.95^CI: lower and upper bound of the 95% confidence interval of hazard ratio; _MC_χ^2^: values of the Mantel–Cox chi-square test and the corresponding probabilities; _TW_χ^2^: values of the Tarone–Ware chi-square test and the corresponding probabilities; *N_C_*: number of the censored cases; *N_E_*: number of the cases that experienced the terminal event. * Surgery was classified as radical or non-radical (biopsy or subtotal resection). ** Peak preoperative hemoglobin levels were classified as low (<120 g/L in females, <140 g/L in males) or high (≥120 or ≥140 g/L, respectively).

**Table 3 biomedicines-12-01210-t003:** Multivariate Cox regression analysis of the most significant categorical and continuous factors in a single model.

Factor	β	SE	Wald	*p*	HR	^0.95^CI_LB_	^0.95^CI_UB_
Age(con)	0.032	0.012	7.333	0.007	1.032	1.009	1.056
Surgery(cat)	−0.874	0.363	5.811	0.016	2.396	1.177	4.877
Epilepsy(cat)	0.599	0.257	5.439	0.020	0.549	0.332	0.909
HIF-1α(con)	0.104	0.053	3.872	0.049	1.110	1.000	1.231
HIF1β(con)	−0.008	0.006	2.095	0.148	0.992	0.980	1.003
Localization(cat)	0.563	0.435	1.679	0.195	1.756	0.749	4.117
KPS(cat)	−0.321	0.311	1.072	0.301	1.379	0.750	2.535
PDGFR(con)	−0.004	0.004	1.027	0.311	0.996	0.988	1.004
IDH1(cat)	−0.406	0.505	0.646	0.422	1.500	0.558	4.033
T-size(cat)	−0.220	0.378	0.338	0.561	1.246	0.594	2.613
Treatment(cat)	−0.163	0.280	0.338	0.561	0.850	0.491	1.471
CD44(con)	0.002	0.005	0.134	0.715	1.002	0.991	1.013
Vimk(con)	0.001	0.006	0.025	0.873	1.001	0.989	1.013

**Abbreviations:** Con: continuous variable; Cat: categorical variable; β: regression coefficient of a variable in the Cox proportional hazard regression model; SE: standard error of β coefficient; Wald: value of the Wald test; *p*: probability of the Wald test; HR: hazard ratio (i.e., exp(β)) for a one-unit change of a variable; ^0.95^CI_LB_ and ^0.95^CI_UB_: lower and upper bound of the 95% confidence interval of HR; KPS: Karnofsky performance status.

## Data Availability

All data mentioned within the manuscript are available upon request to corresponding author.

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
