# Peer review of "Prognostic Factors for Survival in Glioblastoma: A Retrospective Analysis Focused on the Role of Hemoglobin"

_biomedicines, 2024, doi:10.3390/biomedicines12061210_

Round 1
Reviewer 1 Report
Comments and Suggestions for Authors
Dear authors,
First of all, I’d like to give a great congratulation to them for nice and graceful study. Zuzana et al. had a hypothesis that hemoglobin level may be associated with prognosis of glioblastoma, but they failed to show the relationship between Hemoglobin levels and the survival on glioblastoma patients. However, they validated their data with illustrating that the meaningful traditional prognostic factors, such as age; performance status; extent of surgery; postoperative chemoradiotherapy; IDH mutations should be associated with prognosis. Especially hypoxic condition of the tumor showing the expression of HIF-1α; HIF-1β had relationship with prognosis of glioblastoma. Other factors which had an association with prognosis can be interpretated by different way. For example, the tumor depth and tumor size can be considered as a factor for direct association with extent of resection, and vimentin can be considered as a factor for illustrating sarcoma component. As a result, their data showed that good validation with traditional literature.
The manuscript was well written in professional view and the approach to the way resolving hypothesis was much scientific. The topic can be attractive to the readers. However, I am afraid that there is no new concept and low originality in presenting study.
Authors had better describe the reason why they consider the hemoglobin level of glioblastoma patient can be potential prognostic marker more detail. Because the hyperemia is one of potential risk factor for cerebrovascular disease, such as stroke. Readers may be curious about the background of the presenting study.
I think that analysis focused on HIF-1α and HIF-1β can be more informative and has better originality rather than hemoglobin which showed negative result in prognosis analysis. Despite there are also many reports illustrating the prognostic role of HIF-1α; HIF-1β in gliomas, it can be better to report positive relationship on prognosis.
I would like to appreciate reading an interesting article. Good Luck.
Author Response
"Please see the attachment."

Reviewer 2 Report
Comments and Suggestions for Authors
Title
Prognostic Factors for Survival in Glioblastoma: 2 A Retrospective Analysis Focused on the Role of Hemoglobin
General Comments
The study aims at producing univariate and multivariate data by correlating age, extent of surgery, expression of HIF-255 1a, and epilepsy. Main findings were that not correlated with survival or HIF-1a ex-257 expression. Overall the manuscript is written well and highlights the novelty in results, discussion and conclusion sections.
Abstract
Ok
Key words
Ok
Introduction
Ok
Materials and Methods
Ok
Results
Ok
Discussion
Ok
Conclusion
Ok
Supplementary material
Not Applicable
Author contributions
Not evaluated
Acknowledgements
Ok
References
Ok
Author Response
Thank you very much for reading the article and for positive evaluation At the begining of our study, the hypothesis was to correlate level of hemoglobin and HIF-1a with survival, but with knowledge of strong clinical prognostic factors, that alter the survival such as age, extent of surgery, epilepsy we diceded to do multivariete analysis. And yes the final result were negative, which encourages the fact that that there are stronger clinical and biomoleculars (IDH mutation) predicting markers.
Reviewer 3 Report
Comments and Suggestions for Authors
I want to express my appreciation for the opportunity to review this manuscript. I find the work to be quite interesting, and I have a few minor queries that I would like to clarify.
Lane 79: Table 1 is here mentioned but there is no table 1 in these pages.
Patient and methods: section 2.2 histochemical analysis: figure 1 lack of a well described didascaly. The figure lacks a scale bar, labels, and a more detailed description. Furthermore, the text mentioned a method for quantifying immunohistochemical staining, but there is no graph or data on the levels of HIF-1a in patients. Finally, Kaplan-Meier analyses on your data regarding HIF-1a expression levels were not presented.
Statistical analysis: The text specified that associations with various variables between hemoglobin and various mutations of different genes were analyzed. These have not been mentioned further in the text, probably because the analysis did not yield significant results. However, it would be more comprehensive, if possible, to include them in Table 2 or in the supplementary materials along with other non-significant parameters (where other non-significant data are already present). Additionally, since the focus of the study is hemoglobin, it would be interesting to display the Kaplan-Meier data and analysis along with the corresponding graph in the supplementary materials.
Results: In table 2 The parameter Hb is present twice, one of which is marked with two asterisks. However, it would be advisable to find a more direct way to explain the difference in what is intended to be represented in the table regarding the same parameter or to present it only once according to one of the two divisions. Moreover, it's not very clear why the subtotal value N for Hb and Hb** is not equal between them.
Informed Consent Statement: "Informed Consent Statement: I" typing errors, leave a space after the colon and remove the bold from the letter I.
Funding: typing errors: “Funding:Therese” Add a space after the colon
Author Response
"Please see the attachment."
